# *Arabidopsis* transcriptome responses to low water potential using high-throughput plate assays

Stephen Gonzalez[1†], Joseph Swift[1*†], Adi Yaaran[2], Jiaying Xu[1], Charlotte Miller[1], Natanella Illouz-Eliaz[1], Joseph R Nery[3], Wolfgang Busch[1], Yotam Zait[2], Joseph R Ecker[1,3,4]*

[1]Plant Biology Laboratory, The Salk Institute for Biological Studies, La Jolla, United States; [2]The Robert H. Smith Institute of Plant Sciences and Genetics in Agriculture, Faculty of Agriculture, Food, and Environment, The Hebrew University of Jerusalem, Rehovot, Israel; [3]Genomic Analysis Laboratory, The Salk Institute for Biological Studies, La Jolla, United States; [4]Howard Hughes Medical Institute, The Salk Institute for Biological Studies, La Jolla, United States

*For correspondence:
jswift@salk.edu (JS);
ecker@salk.edu (JRE)

†These authors contributed equally to this work

Competing interest: The authors declare that no competing interests exist.

**Abstract** Soil-free assays that induce water stress are routinely used to investigate drought responses in the plant *Arabidopsis thaliana*. Due to their ease of use, the research community often relies on polyethylene glycol (PEG), mannitol, and salt (NaCl) treatments to reduce the water potential of agar media, and thus induce drought conditions in the laboratory. However, while these types of stress can create phenotypes that resemble those of water deficit experienced by soil-grown plants, it remains unclear how these treatments compare at the transcriptional level. Here, we demonstrate that these different methods of lowering water potential elicit both shared and distinct transcriptional responses in *Arabidopsis* shoot and root tissue. When we compared these transcriptional responses to those found in *Arabidopsis* roots subject to vermiculite drying, we discovered many genes induced by vermiculite drying were repressed by low water potential treatments on agar plates (and vice versa). Additionally, we also tested another method for lowering water potential of agar media. By increasing the nutrient content and tensile strength of agar, we show the 'hard agar' (HA) treatment can be leveraged as a high-throughput assay to investigate natural variation in *Arabidopsis* growth responses to low water potential.

## eLife assessment

This work critically evaluates several widely-used assays of transcriptional responses to water limitation in Arabidopsis grown on defined agar-solidified media and, finding inconsistent responses in root transcriptome responses, introduces a new 'hard agar' assay with more consistent responses. The work is **valuable** as a simple and alternative experimental system that would enable high-throughput genetic screening (and GWAS) to assess the impacts of environmental perturbations on transcriptional responses in various genetic backgrounds. Within this scope, the work is **solid**, though the debate about whether field-level physiological inferences can be made from such assays remains.

## Introduction

As climate change advances, improving crop drought tolerance will be key for ensuring food security (*Godfray et al., 2010*; *Battisti and Naylor, 2009*). This has led to intense research at the molecular

level to find novel loci and alleles that drive plant responses to drought conditions. Such investigations benefit from simple assays that can reproduce drought phenotypes at both the physiological and molecular levels. While some researchers use soil-based assays, these are cumbersome. For example, extracting intact root systems from the soil is difficult, and reproducing the rate at which water evaporates from the soil can be challenging (*Dubois and Inzé, 2020*). In light of this, chemical agents such as polyethylene glycol (PEG), mannitol, or salt (NaCl) are often employed to induce drought stress. When present in aqueous or agar media, they allow precise and dose-dependent control of water potential (*Claeys et al., 2014*; *Hohl and Schopfer, 1991*). When exposed to these media types, plants exhibit the hallmarks of drought physiology, such as reduced growth rate, reduced stomatal conductance, and increased leaf senescence (*Claeys et al., 2014*; *Munns, 2002*; *Jisha and Puthur, 2014*). While each of these methods lower water potential and thus induce a drought stress, each method exerts additional and distinct effects. For example, NaCl not only induces osmotic stress, but can cause salt toxicity (*Munns, 2002*). Since it is not metabolized by most plants mannitol is considered less toxic (*Dubois and Inzé, 2020*), however evidence suggests it may act as a signaling molecule (*Hohl and Schopfer, 1991*; *Trontin et al., 2014*). Since both NaCl and mannitol can enter the pores of plant cell walls (*Verslues et al., 2006*; *Juenger and Verslues, 2023*), they can induce plasmolysis, a process that does not typically occur under mild water deficit (*Verslues et al., 2006*). Due to its higher molecular weight, PEG treatment avoids this, and instead causes cytorrhysis (*Verslues et al., 2006*), a physiology more common under drought settings (*Juenger and Verslues, 2023*; *van der Weele et al., 2000*).

The unique impacts PEG, mannitol, and NaCl have on plant physiology may also extend to the level of gene expression. Indeed, a broad spectrum of transcriptional changes are documented in response to low water potential, which may be attributed to the specific method employed (*Claeys et al., 2014*; *Zeller et al., 2009*; *Kreps et al., 2002*). Here, we examine the transcriptional responses to PEG, mannitol, and NaCl in *Arabidopsis*, and compare these responses to those elicited when plants are exposed to vermiculite drying. Furthermore, we explore a new approach for reducing water potential.

## Comparing differential gene expression responses elicited by PEG, mannitol, and NaCl treatment to vermiculite drying

To understand the impact PEG, mannitol, and NaCl treatment have on gene expression, we first tested their physiological effects across a range of doses. To this end, we grew *Arabidopsis* seedlings on agar plates supplemented with Linsmaier & Skoog (LS) nutrients for 14 days on three different doses of each stress type. Dose ranges were chosen based on published literature, and ranged from mild to severe stress levels (*Dubois and Inzé, 2020*; *Claeys et al., 2014*; *van der Weele et al., 2000*). As the dose of each stress type increased, the media's water potential significantly decreased in a dose-dependent manner (Pearson, $p < 8.5 \times 10^{-5}$). Across the doses tested, we found that each stress type's impact on water potential was not statistically different from one other (ANCOVA post hoc, $p > 0.05$). In response to these treatments, we found shoot biomass significantly decreased in a dose-dependent manner (Pearson, $p < 2 \times 10^{-6}$) (*Figure 1A–C*, *Figure 1—figure supplement 1*, *Supplementary file 1*).

Genes that change their expression in response to an environmental signal often do so in a dose-responsive manner (*Claeys et al., 2014*; *Swift et al., 2020*). In light of this, we sought to discover genes whose expression was dose-responsive to the amount of PEG, mannitol, or NaCl applied. By identifying genes that were stress responsive across a range of doses, we ensured such genes responded to and were directional with the stress as a whole, and not induced or repressed at an individual dose. Taking this approach, we sequenced root and shoot bulk transcriptomes by RNA-seq, and associated each gene's expression with the dose of stress with a linear model. To ensure we captured steady-state differences in gene expression, and avoided those that were transient, we sequenced root and shoot transcriptome profiles after 14 days of stress exposure (*Dubois and Inzé, 2020*; *Munns, 2002*; *Nikonorova et al., 2018*). By these means, we found hundreds of genes that were dose-responsive to each treatment within root and shoot tissue (*Figure 1E–H*, *Figure 1—figure supplement 2*, *Supplementary file 2*) (adj. $p < 0.05$). We found that a portion of these dose-responsive genes were shared across treatments, suggesting a common response to low water potential (*Figure 1D*, *Figure 1—figure supplement 3*). Conversely, we also found a portion of dose-responsive genes were unique to each stress type.

Next, we wanted to compare these different methods of lowering water potential to a pot-based water deficit assay. To perform this experiment, we subjected mature *Arabidopsis* plants grown in

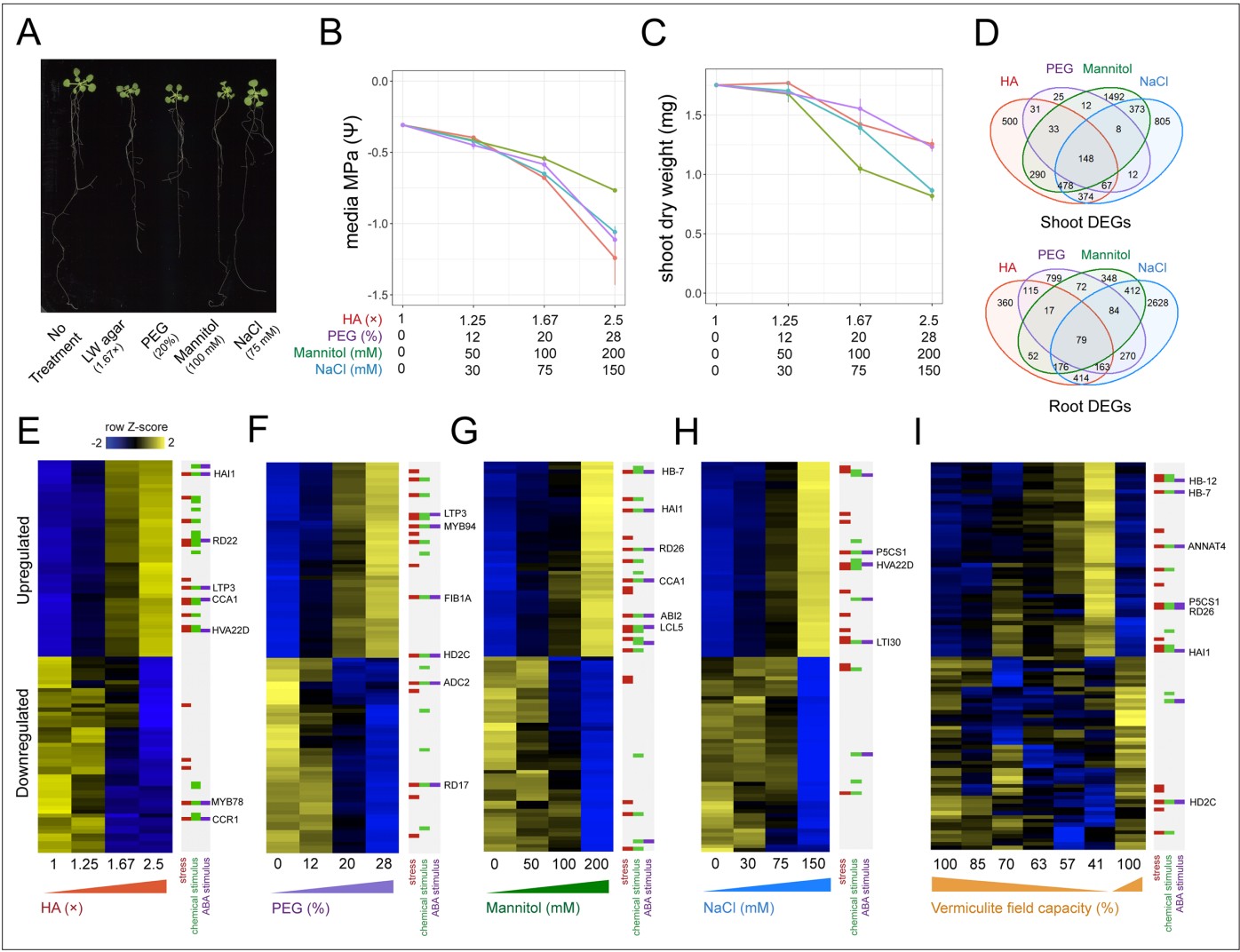

**Figure 1.** Benchmarking the impact different stress assays have on *Arabidopsis* gene expression. (**A**) 22-day-old *Arabidopsis* growth on plates under either 1.67× hard agar (HA), 20% polyethylene glycol (PEG), 100 mM mannitol, or 75 mM NaCl treatments. (**B**) Water potential measurements of treatment media (n=3–4). (**C**) Dry weight of 22-day-old *Arabidopsis* seedlings under different doses of each stress treatment (n=11–12). (**D**) Number and intersect of differentially expressed genes (DEGs) that are dose-responsive to each stress treatment within root and shoot tissue. (**E–I**) Heatmaps displaying the top 50 most significant upregulated or downregulated genes in response to (**E**) HA, (**F**) PEG, (**G**) mannitol, (**H**) NaCl, and (**I**) vermiculite drying in the *Arabidopsis* root (n=2–3 biological replicates). Key genes and membership of Gene Ontology (GO) Terms for 'response to stress', 'response to chemical stimulus', or 'response to ABA stimulus' are indicated. ABA, abscisic acid.

The online version of this article includes the following figure supplement(s) for figure 1:

**Figure supplement 1.** Plant growth responses to stress assays.

**Figure supplement 2.** Shoot gene expression responses to each stress assay are dose-responsive.

**Figure supplement 3.** Overlapping differentially expressed genes (DEGs) responsive to different assay types.

**Figure supplement 4.** Treating vermiculite-grown *Arabidopsis* plants to mild drought stress.

pots on vermiculite supplemented with LS media to a mild water stress by withholding water for 5 days. During this period, field capacity (FC) reduced from 100% to 41%. This treatment led to a reduction in plant biomass (t-test, p=1.8 × 10⁻³), as well as seed yield (t-test, p=1.2 × 10⁻⁴), but did not induce visible signs of senescence or wilting (*Figure 1—figure supplement 4*, *Supplementary file 3*). We assayed root and shoot gene expression responses each day during water loss by RNA-seq. We observed a dose-dependent relationship between a decrease in FC and gene expression responses in both roots and shoots, identifying 1949 differentially expressed genes in roots and 1792 in shoots (DESeq, adj. p<0.01) (*Figure 1I*, *Figure 1—figure supplement 2*, *Supplementary file 2*). We ensured

these genes' expression patterns recovered upon rewatering (*Figure 1I*). We note that while vermiculite has greater aeration than soil, we found that the genes differentially expressed in roots in response to vermiculite drying largely agreed with a previous report detailing transcriptional responses to soil drying (*Lozano-Elena et al., 2022*; *Figure 2—figure supplement 1*). We also found differentially expressed genes responsive to vermiculite drying agreed with those responsive to transient treatment with abscisic acid (ABA), a stress hormone whose levels rise in response to water deficit (*Claeys et al., 2014*; *Figure 2—figure supplement 2*).

To assess how PEG, mannitol, and NaCl treatments compared to the vermiculite drying response described above, we overlapped genes found differentially expressed in each experiment. For shoot tissue, we found genes that were differentially expressed during vermiculite drying overlapped significantly with genes that were differentially expressed by either PEG, mannitol, and NaCl treatments (Fisher's exact test, adj. p<0.05). Additionally, there was 88–99% directional agreement within these overlaps, indicating that genes induced or repressed by vermiculite drying were similarly induced or repressed by low water potential treatments on agar (*Figure 2A and B*). Along these lines, across all conditions we saw differential expression of the desiccation associated genes *RESPONSE TO DESSICATION 20;29B (RD20;29B)* (*Msanne et al., 2011*; *Takahashi et al., 2000*), the osmo-protectant gene *DELTA1-PYRROLINE-5-CARBOXYLATE SYNTHASE 1* (*P5CS1*), and ABA signaling and biosynthesis genes *HOMEOBOX 7* (HB7) (*Valdés et al., 2012*), and *NINE-CIS-EPOXYCAROTENOID DIOXYGENASE* (*NCED3*) (*Tan et al., 2003*; *Figure 2—figure supplement 3*). We note that while we observed agreement in the direction of gene expression across assays, there were differences in the amplitude of gene expression (*Figure 2—figure supplement 3*). This may be due to confounding factors, such as differences in the ranges of water potential tested (*Figure 1B*), or through comparing seedlings grown on plates with mature *Arabidopsis* plants grown on vermiculite.

In root tissue, we found greater variability in transcriptomic responses to the different methods of lowering water potential. In particular, we found a number of genes that were upregulated during vermiculite drying were downregulated by PEG, mannitol, and NaCl treatments (and vice versa) (*Figure 2A*). This trend persisted when we assessed genes found differentially expressed at each discrete dose of stress (*Figure 2B*). For example, 27% of PEG dose-responsive genes shared the same direction of expression seen in vermiculite drying responses. We note that previously published PEG transcriptome datasets largely agreed with our own (*Figure 2—figure supplement 1*). Such differential regulation in comparison to vermiculite drying is exemplified by the expression of genes such as *HOMEOBOX 12* (*HB12*) (*Valdés et al., 2012*; *Figure 2C*), *GRC2-LIKE 1* (*GCL1*) (*Gao et al., 2007*), and *RESPONSE TO DEHYDRATION* 21 (*RD21*) (*Koizumi et al., 1993*; *Figure 2—figure supplement 3*). We found genes downregulated by PEG are over-represented in the 'monooxygenase activity', and 'oxygen binding' Gene Ontology (GO) Terms ($p<1 \times 10^{-15}$, *Supplementary file 4*). Mannitol and NaCl held a 48% and 57% agreement in gene expression direction with vermiculite drying respectively. Examples of genes that followed this pattern of differential regulation in mannitol and NaCl treatments were *DROUGHT HYPERSENSITIVE 2* (*DRY2*) (*Posé et al., 2009*; *Figure 2D*) and *ROOT HAIR SPECIFIC 18* (RHS18) (*Ponce et al., 2022*). NaCl-responsive GO Terms included a specific downregulation of 'phosphorous metabolic processes' ($p=5.2 \times 10^{-6}$), suggesting that the roots were changing phosphate levels in response to NaCl, a process known to help maintain ion homeostasis (*Miura et al., 2011*). For mannitol, we observed a specific downregulation of 'cell wall organization or biogenesis' and 'microtubule-based processes' ($p<7.8 \times 10^{-3}$) (*Supplementary file 4*).

## Examining differential gene expression responses to 'hard agar' (HA) treatment

In addition to examining PEG, mannitol, and NaCl transcriptional responses, we also tested a new way of lowering water potential on an agar plate. We hypothesized that we could induce stress by increasing both the agar and nutrient concentration. We called this media 'hard agar' (HA), and by testing three different doses (1.25×, 1.67×, and 2.5× fold increase in both agar and LS concentration, where 1.0× is 2% agar and 1× LS), found that it limited plant shoot dry weight and media water potential in a similar way to PEG, mannitol, and NaCl treatment (*Figure 1A–C*). Additionally, we found that HA treatment limited *Arabidopsis* primary root growth rate, shoot water potential, and photosynthesis efficiency (*Figure 2—figure supplement 4*). At the molecular level, RNA-seq revealed 1376 and 1921 genes that were dose-responsive to the level of HA stress in roots and shoots respectively (*Figure 1E*,

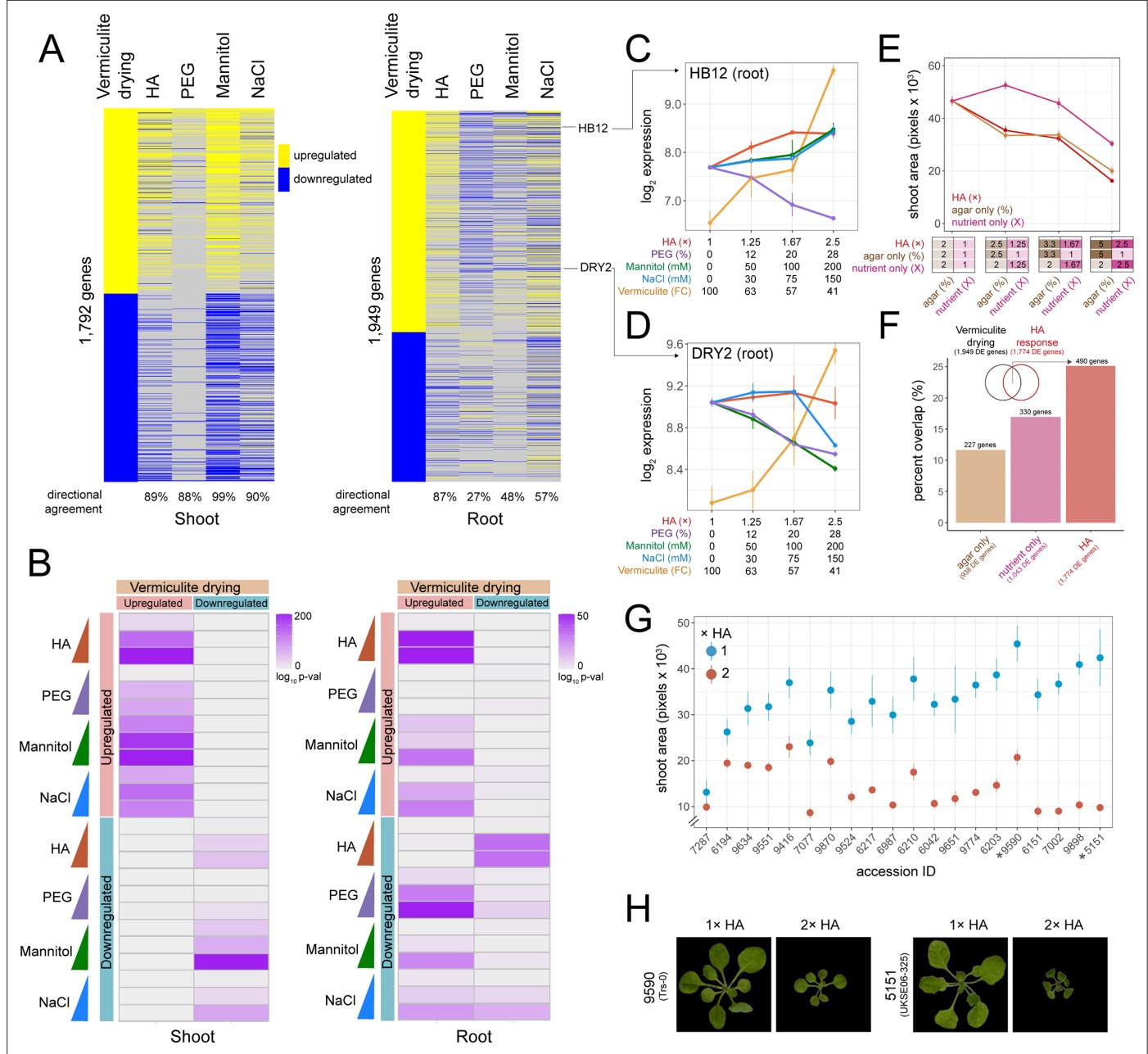

**Figure 2.** Comparing hard agar (HA), polyethylene glycol (PEG), mannitol, and NaCl gene expression responses to vermiculite drying. (**A**) Heatmap displaying genes differentially expressed in response to vermiculite drying in shoot or root tissue compared to their dose-responsive expression within each plate-based assay. Level of 'directional agreement' (i.e. differentially expressed in the same direction) found within each assay reported. (**B**) Overlap analysis of genes found differentially expressed due to vermiculite drying, compared to those found differentially expressed within each dose of PEG, mannitol, NaCl, or HA assays in both shoot and root (Fisher's exact test, adj. p<0.05). (**C–D**) Expression patterns of *HOMEOBOX12* (*HB12*) and *DROUGHT HYPERSENSITIVE 2* (*DRY2*) across each assay in root tissue (n=2-3). (**E**) Shoot area of seedlings grown under increasing doses of HA, agar, or nutrient concentrations (n=19). (**F**) Number and percent overlap of genes found differentially expressed in response to increasing doses of HA, agar, or nutrient concentrations with those differentially expressed in response to vermiculite drying. (**G**) Total shoot area of *Arabidopsis* accessions grown under either 1× or 2× HA treatment (n=5–12). (**H**) Images of *Arabidopsis Trs-0* or *UKSE06-325* accessions grown on either 1× or 2× HA treatment.

The online version of this article includes the following figure supplement(s) for figure 2:

**Figure supplement 1.** Comparing gene expression responses to vermiculite drying and polyethylene glycol (PEG) treatment with previous studies.

**Figure supplement 2.** Comparing abscisic acid (ABA)-induced differential expression to vermiculite drying and hard agar (HA)-induced gene expression patterns.

**Figure supplement 3.** Gene expression profiles of individual genes.

*Figure 2 continued on next page*

*Figure 2 continued*

**Figure supplement 4.** Physiological measurements of *Arabidopsis* seedlings in response to hard agar (HA) treatment.

**Figure supplement 5.** Comparing the separate effects of nutrient concentration and agar concentration on seedling growth.

**Figure supplement 6.** Associating hard agar's (HA) impact on shoot size with plant fitness.

**Figure supplement 7.** The volume of hard agar (HA) has minimal impact on gene expression.

*Figure 1—figure supplement 2*). We found that these gene expression responses overlapped significantly with those found differentially expressed in response to vermiculite drying (Fisher's exact test, $p < 1 \times 10^{-32}$, 87% directional agreement) (*Figure 2A and B*). HA's impact can be seen in the gene expression responses of *HB12* (*Figure 2C*), *GCL1,* and *RD21* (*Figure 2—figure supplement 3*).

An increase in nutrient concentration can induce salt-like stress while increasing agar concentration will increase tensile stress (*Verger et al., 2018*). We tested each of these variables separately to understand the role each played in eliciting the gene expression responses found in the HA assay. To do this, we repeated our HA dose experiment, but now increasing only the concentration of LS nutrients (1×, 1.25×, 1.67×, and 2.5×) or the concentration of agar (2%, 2.5%, 3.3%, and 5%) (*Figure 2E*). We found a significant decrease in shoot area size in response to an increase in nutrient concentration (Pearson $p = 1.7 \times 10^{-7}$) and agar concentration (Pearson $p = 3.9 \times 10^{-13}$), where the latter more closely phenocopied the effect of HA (*Figure 2E*, *Figure 2—figure supplement 5*, *Supplementary file 5*). Since the increase in nutrient concentration alone was responsible for changing media water potential, the phenotypic response to increased agar concentration was not in response to a lower water potential (*Figure 2—figure supplement 5*). Next, we examined the transcriptional responses underlying nutrient and agar responses by sequencing root tissue across each dose tested. Through linear modeling, we found 1043 genes and 938 genes that were dose-responsive to nutrient or agar concentration, respectively. Then, we investigated how these genes compared to those found in vermiculite drying responses. We found that genes differentially expressed in response to an increase in agar or nutrient concentration overlapped 12% and 17% of vermiculite drying responsive gene expression respectively (permutation test, $p < 0.05$) (*Figure 2F*, *Figure 2—figure supplement 5*, *Supplementary file 6*). However, we found genes differentially expressed in response to HA treatment led to a higher overlap (26 %), suggesting that both nutrient and agar concentration contribute to the similarity between HA treatment and vermiculite drying.

Finally, we tested if our HA assay was sensitive enough to detect phenotypic variability. To achieve this, we grew 20 different *Arabidopsis* ecotypes on 2× HA (4% agar, 2× LS), where ecotypes were selected from a previous drought study that assessed fitness in a common garden experiment (*Exposito-Alonso et al., 2019*). By comparing the total shoot area after 3 weeks of growth, we found that our assay revealed variability in shoot growth responses (*Figure 2G and H*, *Supplementary file 7*). Furthermore, we found that the greater the impact HA had on reducing an accession's relative shoot size, the better the accession's fitness was, as measured under field conditions (*Exposito-Alonso et al., 2019*) (Spearman $p = 0.04$, *Figure 2—figure supplement 6*). This is likely because smaller shoot systems have a better chance of survival and reproduction (*Skirycz et al., 2011*). This suggests that our assay may be useful for screening for novel drought-associated loci among a wider group of accessions or mutants.

In summary, our investigation has assessed the shared and unique impacts of agar-based low water potential treatments on gene expression. We also compared these effects with the expression patterns observed during vermiculite drying. We found each plate-based assay generated similar responses in shoot tissue, but more varied responses in root tissue. We note that our comparative analysis focuses largely on transcriptomic responses in *Arabidopsis*. We suggest investigating gene expression responses in other species as future work. Here, we also introduce another method for lowering water potential. By increasing nutrient and agar concentration, our HA approach also induced gene expression responses comparable to vermiculite drying. We describe how to make this media within the Materials and methods.

## Materials and methods

### HA stress assay

*Arabidopsis* seedlings were grown on vertical plates for 8 days under short-day conditions (8 hr light, 21°C, 150 µmoles light) on agar media (1× LS (Cassion LSP03) media, 1% sucrose, 2% agar, pH 5.7). We note LS media is identical to Murashige & Skoog media in inorganic salt content, but lacks glycine, nicotinic acid, and pyridoxine HCl. After 8 days, plants were transferred to HA plates. The 1.0× plate consisted of 2% and 1× LS media, with no sucrose (pH 5.7) at a final volume of 75 mL. Subsequent doses of increased nutrient and agar concentration (1.25×, 1.67×, and 2.5× fold increase) were made by preparing the same media but reducing the amount of water present. For example, the 1.25×, treatment plate contained 60 mL of 2.5% agar and 1.25× LS media. We note that the volume of HA itself has minimal impact gene expression responses (*Figure 2—figure supplement 7*). On day 14, 2 hr after subjective dawn, shoot and root samples were flash-frozen (6 plants per replicate). In total, we collected 16 samples for RNA-seq analysis (2 organs, 2–3 biological replicates, 3 treatment levels). We also collected a non-treated control set (2 biological replicates).

To test different *Arabidopsis* accessions on HA, plants were sown on either 1× or 2× treatments as described above, however supplemented with 0.5% or 1% sucrose respectively to encourage germination. Seedlings were grown for 3 weeks under short-day conditions in before imaging plates in duplicate (n=2–5 plants per plate) (*Supplementary file 7*). Shoot area was calculated from images using Plant Growth Tracker (GitHub - https://github.com/jiayinghsu/plant-growth-tracker; *Xu, 2022*).

### Vermiculite drying assay

*Arabidopsis* seedlings were grown on vertical plates for 17 days under short-day conditions (8 hr light, 21°C, 150 µmoles light) on agar media (1× LS, 1% sucrose, 2% agar, pH 5.7), before transfer to vermiculite saturated with 0.75× LS media. We note at the timing of transfer lateral root formation had begun. Plants were then grown on vermiculite at 100% field capacity (FC) for 12 days (8 hr light, 21°C, 150 µmoles light). On the 13th day, the first time point was sampled (4.5 hr after subjective dawn) where tissue was flash-frozen in liquid nitrogen. After this, excess aqueous solution was drained from each pot, and then each pot was calibrated to 1× FC. Plant tissue was harvested each day on subsequent days at the same time of day. Each day, pots were weighed to measure extent of evaporation. By these means, FC was measured (*Figure 1—figure supplement 4*). After the 5th day sample was taken, water was re-added to the remaining pots to an excess of 1× FC. ~15 plants were sampled per time point. In total, we harvested 78 tissue samples for RNA-seq (3 biological replicates, 2 organ types, 7 days, 2 treatments). Plants were then left to grow under long-day conditions until flowering. Seeds were harvested, dried, and weighed (n=50 plants per treatment).

### PEG stress assay

*Arabidopsis* seedlings were grown on vertical plates for 8 days under short-day conditions (8 hr light, 21°C, 150 µmoles light) on agar media (1× LS, 1% sucrose, 2% agar, pH 5.7), before transfer to PEG media of varying concentrations. PEG media plates were prepared by dissolving crystalline 6000 MW PEG into freshly autoclaved 1× LS media pH 5.7 and pouring 50 mL of PEG media solution onto 1× LS, 2% agar, media plates (pH 5.7), letting the PEG solution diffuse into the solid media overnight, then pouring off excess and transferring seedlings to PEG infused media plates as described in *van der Weele et al., 2000*. Plants were grown under three different treatments (12%, 20%, and 28% PEG solution wt/vol) for 14 days. On day 14, 2 hr after subjective dawn, shoot and root samples were flash-frozen (6 plants per replicate). In total, we collected 16 samples for RNA-seq analysis (2 organs, 2–3 biological replicates, 3 treatment levels).

### Mannitol and NaCl osmotic stress assays

*Arabidopsis* seedlings were grown on vertical plates for 8 days under short-day conditions (8 hr light, 21°C, 150 µmoles light) on agar media (1× LS, 1% sucrose, 2% agar, pH 5.7), before transfer to either mannitol or salt (NaCl) media of varying concentrations. Mannitol and NaCl media plates were prepared by adding respective volume of stock solution to 1× LS, 2% agar, pH 5.7 media before autoclaving. Plants were grown under three different treatments of mannitol or NaCl (50 mM, 100 mM, and 200 mM for mannitol, 30 mM, 75 mM, and 150 mM for NaCl) for 14 days. On day 14, 2 hr after subjective dawn, shoot and root samples were flash-frozen (6 plants per replicate). In total, for either

mannitol or NaCl treatment experiments, we collected 18 samples for RNA-seq analysis (2 organs, 3 biological replicates, 3 treatment levels).

## ABA exogenous treatment assay

*Arabidopsis* seedlings were grown on vertical plates for 8 days under short-day conditions (8 hr light, 21°C, 150 µmoles light) on agar media (1× LS, 1% sucrose, 2% agar, pH 5.7), before transfer to 1× LS, 2% agar, pH 5.7 control media and grown for 14 days. On day 14, ABA solutions of 1 µM, 5 µM, and 10 µM were prepared from 10 mM ABA dissolved in ethanol stock, as well as a mock treatment solution containing 0.1% ethanol concentration. 30 min after subjective dawn, 15 mL of each solution was dispersed onto the roots of the seedlings. After 1 min of treatment, the ABA solution was removed from the plates, and the plates returned to the growth chamber. 2 hr after subjective dawn, shoot and root samples were flash-frozen (6 plants per replicate). In total, we collected 8 samples for RNA-seq analysis (root tissue only, 2 biological replicates, 4 conditions).

## Osmotic potential measurements

The water potential of media was determined considering it equivalent to the osmotic potential ($\Psi$s). Osmotic potential was measured using a vapor pressure osmometer (Model 5600, ELITech Group; Puteaux, France). Readings were taken from melted agar media constituted with the particular stress type. Osmolality readings for each sample obtained were converted to megapascals (MPa) using the equation $\Psi s = -CRT$, where C is the molar concentration, R is the universal gas constant, T is the temperature in Kelvin. We note that to measure the water potential of PEG treatment media, we infiltrated the PEG solution into plates as described above, and then melted the PEG-infiltrated agar for measurement with the osmometer. We assessed the osmotic potential of shoot tissue 2 weeks after transplanting the seedlings to HA media. After immersion in liquid nitrogen 3 shoots were placed into 0.5 mL tubes and centrifuged to extract the tissue sap. The osmotic potential ($\Psi$s) of the extracted sap was determined using a vapor pressure osmometer.

## Chlorophyll fluorescence measurements

Chlorophyll fluorescence was assessed in eight seedlings of each plate using the Walz PAM IMAGING PAM M-series IMAG-K7 (MAXI) fluorometer. For every experiment, leaves were pre-conditioned in the dark for 1 hr. The maximum quantum yield of photosystem II (PSII) ($F_v/F_m$) was calculated using the formula:

$$F_v/F_m = (F_m - F_o)/F_m$$

where $F_v$ is the variable fluorescence, $F_m$ is the maximal fluorescence following 1 hr of dark adaptation, and $F_0$ is the minimal fluorescence level of a dark-adapted leaf when all PSII reaction centers are open.

## Root growth rate measurements

*Arabidopsis* seedlings were grown on vertical plates for 8 days under short-day conditions (8 hr light, 21°C, 150 µmoles light) on agar media (1× LS, 1% sucrose, 2% agar), before transfer to 2.5× HA treatment plates as described above. Root images were acquired every 2 days for a total of 8 days using scanners. Primary root length, defined as the length (scaled to cm) from hypocotyl base to root tip, was quantified using ImageJ. For each treatment we screened 4 plates, with each plate holding 4 individual plants.

## RNA extraction and library preparation

Plant tissue was crushed using the TissueLyser (Agilent) and RNA extracted using RNeasy Mini Kit (QIAGEN). Number of biological replicates per library ranged between RNA quality was assessed using TapeStation High Sensitivity RNA assay (Agilent). 0.5–1 µg of total RNA proceeded to library preparation, where libraries were prepared using TruSeq stranded mRNA kit (Illumina). Resulting libraries were sequenced on the NovaSeq 6000 (Illumina) with 2×150 bp paired-end read chemistry. Read sequences were aligned to the *Arabidopsis* TAIR10 genome using HISAT2 (*Kim et al., 2019*), and gene counts called using HT-seq (*Anders et al., 2015*), by relying on Araport11 annotation (*Cheng et al., 2017*). Normalized counts can be found in *Supplementary file 2*. For each organ, libraries from all experiments were normalized together before calling differential expression.

## Statistical analysis

To detect differential expression in our drought assay on vermiculite, we called differential expression using a linear model using the DESeq2 LRT function to associate a change in FC with change in gene expression. The same statistical approach was used to associate a change in a gene's expression to changes in dose of HA, PEG, mannitol, and NaCl, as well as changes in agar concentration, nutrient concentration, and volume of agar used. Resulting model p-values were adjusted to account for false discovery (p-value<0.05). The complete list of differentially expressed genes for each experiment can be found in *Supplementary file 2* and *Supplementary file 6*. Pairwise differential gene expression was called using DESeq2 (*Love et al., 2014*). Specifically, for plate-based assays, we called differential expression by comparing the control treatment to each treatment dose, using an adjusted p-value threshold of 0.05. Overlap analyses were performed using Fisher's exact tests, with an adjusted p-value threshold of 0.05. The background for these intersects was all expressed genes within the respective organ. Permutation tests and GO Term enrichment analyses were performed in VirtualPlant (*Katari et al., 2010*), with all expressed genes within the respective organ used as background.

## Acknowledgements

We thank Renee Garza for critical reading of the manuscript. JS is an Open Philanthropy awardee of Life Science Research Foundation, as well as recipient of the Pratt Industries American-Australian Association Scholarship. JRE is an Investigator of the Howard Hughes Medical Institute.

## Additional information

### Funding

| Funder | Grant reference number | Author |
| --- | --- | --- |
| Howard Hughes Medical Institute | | Joseph R Ecker |

The funders had no role in study design, data collection and interpretation, or the decision to submit the work for publication.

### Author contributions

Stephen Gonzalez, Conceptualization, Data curation, Formal analysis, Validation, Investigation, Methodology, Writing - original draft, Project administration, Writing - review and editing; Joseph Swift, Conceptualization, Data curation, Formal analysis, Supervision, Validation, Investigation, Visualization, Methodology, Writing - original draft, Project administration, Writing - review and editing; Adi Yaaran, Data curation, Investigation, Methodology; Jiaying Xu, Natanella Illouz-Eliaz, Software, Methodology; Charlotte Miller, Investigation, Methodology; Joseph R Nery, Data curation; Wolfgang Busch, Yotam Zait, Supervision, Methodology; Joseph R Ecker, Conceptualization, Supervision, Funding acquisition, Investigation, Methodology

### Author ORCIDs

Stephen Gonzalez http://orcid.org/0009-0005-8339-9911
Joseph Swift http://orcid.org/0000-0001-9559-1699
Joseph R Ecker http://orcid.org/0000-0001-5799-5895

Reviewer #2 (Public Review): https://doi.org/10.7554/eLife.84747.3.sa1
Reviewer #3 (Public Review): https://doi.org/10.7554/eLife.84747.3.sa2
Author response https://doi.org/10.7554/eLife.84747.3.sa3

## Additional files

### Supplementary files

• Supplementary file 1. Plant physiological measurements.

- Supplementary file 2. Differentially expressed genes and normalized counts in hard agar (HA), polyethylene glycol (PEG), mannitol, NaCl, or vermiculite drying experiments.
- Supplementary file 3. Vermiculite drying assay measurements.
- Supplementary file 4. Gene Ontology (GO) Term enrichment of differentially expressed genes.
- Supplementary file 5. Shoot area of seedlings grown under different agar and nutrient concentrations.
- Supplementary file 6. Differentially expressed genes and normalized counts in response to changes in nutrient or agar concentration.
- Supplementary file 7. Shoot area of different *Arabidopsis* accessions grown on hard agar (HA) media.
- MDAR checklist

## Data availability

Raw sequencing data can be found at the National Center for Biotechnology Information Sequence Read Archive (accession number PRJNA904764). Normalized read counts and raw phenotypic datasets can be found in the Supplementary Material.

The following dataset was generated:

| Author(s) | Year | Dataset title | Dataset URL | Database and Identifier |
|---|---|---|---|---|
| Gonzalez S, Swift J, Ecker J | 2022 | Mimicking genuine drought responses using a high throughput plate assay | https://www.ncbi.nlm.nih.gov/bioproject/PRJNA904764 | NCBI BioProject, PRJNA904764 |

The following previously published dataset was used:

| Author(s) | Year | Dataset title | Dataset URL | Database and Identifier |
|---|---|---|---|---|
| Lozano-Elena F, Fàbregas N, Coleto-Alcudia V, Caño-Delgado AI | 2018 | Transcriptomic study of Arabidopsis roots overexpressing the brassinosteroid receptor BRL3, in control conditions and under severe drought | https://www.ncbi.nlm.nih.gov/geo/query/acc.cgi?acc=GSE119382 | NCBI Gene Expression Omnibus, GSE119382 |

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
