## [Editor Report · eLife assessment]

This work critically evaluates several widely-used assays of transcriptional responses to water limitation in Arabidopsis grown on defined agar-solidified media and, finding inconsistent responses in root transcriptome responses, introduces a new 'hard agar' assay with more consistent responses. The work is **valuable** as a simple and alternative experimental system that would enable high-throughput genetic screening (and GWAS) to assess the impacts of environmental perturbations on transcriptional responses in various genetic backgrounds. Within this scope, the work is **solid**, though the debate about whether field-level physiological inferences can be made from such assays remains.

---

## [Referee Report · Reviewer #2 (Public Review)]

This manuscript describes new methodology to study low water potential (drought) stress responses in agar plates. They devote considerable effort in comparing transcriptome data among various previously published experimental systems, examining how different approaches of reducing water potential impact the Arabidopsis root and shoot transcriptome. Each method purported to reduce water potential in plate-grown seedlings has a different effect on Arabidopsis root transcriptome responses, which is problematic for the field. In this reviewer's view, differences in transcriptome are not as important, and often not as informative as measurement of physiological parameters, which they do very little of in their study.

The focus on transcriptome data to the almost complete exclusion of other types of data is a symptom of a broader over-emphasis on the transcriptome that is quite prevalent in plant science now. We measure transcriptomes because we can, not because it is inherently the most informative thing to do. The important thing is protein amount, and even more so protein activity/function, which we know has an imperfect, at best, correlation with transcript level. This reviewer acknowledges that using Arabidopsis transcriptomics is a commonly employed method, and as such, the outcomes of this study will hold value for a broad audience, even if largely as a cautionary tale. If transcriptomics is used to identify candidate genes for future investigations, an approach that has had some success, then appropriate cautions should be taken in translating expectations about gene, protein, and phenotypic responses in field conditions.

---

## [Referee Report · Reviewer #3 (Public Review)]

This work compares transcriptional responses of shoots and roots harvested from four plate-based assays that aim to simulate drought and from plants subjected to water deficit in pots using the model plant *Arabidopsis thaliana* with the goal to select a plate-based assay that best recapitulates transcriptional changes that are observed during water-deficit in pots. For the plate-based assays polyethylene glycol (PEG), mannitol, and sodium chloride (salt) treatments were used as well as a 'hard agar' assay which was newly developed by the authors. In the 'hard agar' assay, less water was added to the solid components of the media leading to an increase in agar strength and nutrient concentration. Plants in pots were grown on vermiculite with the same nutrient mix as used in the plates and drought was induced by withholding watering for five days.

The authors observed a good directional agreement of differential expressed genes for shoots between the plate assays on the vermiculite drying experiment. However, less directional agreement was observed for differential expressed genes of roots, except for their newly developed 'hard agar' assay which had good directional agreement. Testing whether the increase in agar strength or more concentrated nutrients are attributed to this, they found that both factors contributed to the effect of the 'hard agar'. Arabidopsis ecotypes that showed a stronger reduction in shoot size when grown on the 'hard agar' tended to have a lower fitness according to an external study which may indicate that the 'hard agar' assay simulates physiological relevant conditions.

The work highlights that transcriptional responses for simulated drought on plates and drought caused by water deficit are highly variable and dependent on the tissues that are observed. The authors demonstrate that transcriptomics can be used to select a suitable plate assay that most closely recapitulates drought through water deficit for plants grown in pots. Interestingly their newly developed 'hard agar' assay provides an alternative to traditional plate-based assays with improved directional agreement of differential expressed genes in roots in comparison to plants experiencing water deficit in vermiculite. It is promising that the impact of 'hard agar' on the shoot size of 20 diverse Arabidopsis accessions shows some association with plant fitness under drought in the field. Their methodology could be powerful in identifying a better substitute for plate-based high-throughput drought assays that have an emphasis on gene expression changes.

---

## [Author Response]

The following is the authors’ response to the previous reviews

**eLife assessment**
This work is an attempt to establish conditions that accurately and efficiently mimic a drought response in Arabidopsis grown on defined agar-solidified media - an admirable goal as a reliable experimental system is key to conducting successful low water potential experiments and would enable high-throughput genetic screening (and GWAS) to assess the impacts of environmental perturbations on various genetic backgrounds. The authors compare transcriptome patterns of plant subjected to water limitation imposed with different experimental systems. The work is valuable in that it lays out the challenges of such an endeavor and points out shortcomings of previous attempts. There was concern, however, that a purely gene expression-based approach may not provide sufficient physiologically relevant information about plant responses to drought, and therefore, despite improvements from a previous version, the new methodology championed by this work remains inadequate.

Molecular biologists who study drought stress must make choices about which assays to use in their investigation. Serious resources and effort are put into their endeavor, and choice of assay matters. Our manuscript’s goal was largely practical: to guide molecular biologists employing transcriptomics in their choice of drought stress assay, and thus help ensure their work will discover transcriptional signatures of importance, and not those that may be an artifact from lowering water potential using chemical agents on agar plates.

We examine how different approaches of reducing water potential impact the *Arabidopsis* root and shoot transcriptome. Our manuscript shows that each method of reducing water potential has a different effect on *Arabidopsis* root transcriptome responses. We acknowledge that drought stress induces a complex physiological response, and can vary depending on the method used. However, by comparing across assays, we find instances where a gene is downregulated by low water potential in one assay, and upregulated by low water potential in another assay. We feel it is only natural to question why this could be, and to hypothesize that it may be caused by secondary effects caused by the way low water potential is imposed. We note that comparative transcriptomics has been a standard approach for decades. We take it as the reviewer’s opinion that it may not be insightful, but it does not factually impact our findings.

**Reviewer #2 (Public Review):**
This manuscript purports to develop a new system to study low water potential (drought) stress responses in agar plates. They make numerous problematic comparisons among transcriptome datasets, particularly to transcriptome data from a vermiculite drying experiment which they inappropriately present as representing an authentic "drought response" to the exclusion of all other data. For some reason, which the reviewer cannot fully understand, the authors seem intent on asserting the superiority of their experimental system to all others. They do not succeed in this and such an effort is ultimately a disservice to the field of drought research as a whole.While they devote considerable effort in comparing transcriptome data among various experimental systems, the potentially more informative experiment at the end of the manuscript of testing growth responses of a number of Arabidopsis accessions is only done for their "LW" system. The focus of this manuscript on transcriptome data to the almost complete exclusion of other types of data which is a symptom of a broader over-emphasis on transcriptome that unfortunately is quite prevalent in plant science now. It is worth reminding that for protein coding genes, which constitute the vast majority of genes, transcriptome data is a proxy measurement. The really important thing is protein amount, and even more so protein activity/function, which we know has an imperfect, at best, correlation with transcript level. We measure transcriptomes because we can, not because it is inherently the most informative thing to do. The author's quixotic quest to see if the transcriptomes of different stress treatments match is of limited value and further diminished by their misleading presentation of one particular transcriptome data set (from their vermiculite drying experiments) as somehow a special data set that everything else must be evaluated against. This study sheds no new light on how to do relevant drought (low water potential) experiments in the lab.Although the reviewer acknowledges that the authors have made some effort to respond to previous comments, the fundamental flaws remain and the present version of this study is little improved from the first submission.

One challenge faced by the drought community is establishing consensus regarding the definition of drought itself. According to the criteria followed by the reviewer, any method leading to a reduction in water potential qualifies as drought stress. However, the findings presented in this manuscript demonstrate that transcriptional responses in roots vary considerably across five different methods of reducing water potential. This indicates that beyond responding to a change in water potential itself, root transcriptomes will also respond to the specific way low water potential is introduced. We believe this variability is of interest to the drought research community.

Of the five methods we explore, we hold the view that the gene expression changes induced by vermiculite drying as the most analogous to the expression signatures *Arabidopsis* would exhibit in response to low water potential in the natural environment. In contrast, we posit that *Arabidopsis* grown on agar plates - where the root system is exposed to air and light, and where water potential is lowered using chemical agents - may contain gene expression signatures plant molecular biologists may not find particularly relevant. However, we acknowledge that this is our opinion, and will make this more explicit on our revised text.

More broadly, we believe that the reviewer’s observation regarding the ‘over-emphasis’ on transcriptomics that is prevalent within the plant science community justifies, rather than diminishes, the work presented here. If transcriptomics is a commonly employed method, then we anticipate that the outcomes of this study will hold value for a broad audience. Such researchers are likely not only using transcriptomics as a proxy measure for protein abundance, as the reviewer suggests, but also because it is one of the more straightforward genomic techniques biologists can use to identify candidate genes that may be chosen for further scrutiny.

**Reviewer #3 (Public Review):**
Comments on revised version:Specific previous criticisms that were addressed are:(1) that gene expression changes were only compared between the highest dose of each stress assay. In the revised version, the authors changed their framework and are now using linear modelling to detect genes that display a dose response to each specific treatment. I agree that this might be a more robust approach to selecting genes that are specific to a certain treatment.(2) that concentrations of PEG, mannitol, NaCl, and the "low water" agar which were chosen are not comparable in regards to their specific osmotic component. I appreciate that the authors measured the osmotic potential of each treatment. It revealed that both PEG and NaCl at their highest concentration had a much more negative osmotic potential compared to the other treatment. The authors claim that using ANCOVA they did not detect any significant differences between the treatments (lines 113, 114). I do believe that ANCOVA is not the appropriate test in this case. ANCOVA has an assumption of linearity, while the dose response between concentration and osmotic potential is non-linear. This is particularly evident for PEG (Steuter AA. Water potential of aqueous polyethylene glycol. Plant Physiol. 1981 Jan;67(1):64-7. doi: 10.1104/pp.67.1.64.). Since the treatments are not the same at the highest level, I think this could have effects on the validity of comparisons by linear model. One approach could be to remove the treatment level with the highest concentration and compare the results or adjust the treatments to the same osmolarity.(3) that only two biological replicates were collected for RNA sequencing which makes it impossible to know how much variance exists between samples. The authors added a third replicate in the revised version for most treatments. However, some treatments still have only two replicates, which cannot be easily seen from the text or the figure. I would prefer that those differences are pointed out.(4) that the original manuscript did not explore what effect the increase of agar and nutrient concentration in the "low water" agar had on water potentials. The authors conducted additional experiments showing that changes in water potential were exclusively caused by changes in the nutrient concentration (Figure 2-figure supplement 5; lines 222-224). However, the increase in agar strength had also some effect on gene expression. While this is not further discussed in the text, I believe this effect of agar on gene expression could be similar to root responses to soil compaction.(5) That the lower volume of media in the "low water" agar could have an effect on plants. The authors compared these effects in Figure 2-figure supplement 7. They claim that "different volumes of LW agar media do not play a significant part in modulating gene expression". While I can see that they detected 313 overlapping DEGs, there were still 146 and 412 non-overlapping DEGs. The heatmap in subpanel E also shows that there were differences in particular in the up-regulated genes. My conclusion would be that the change in volume does play a role and this should be a consideration in the manuscript.

We thank the reviewer for their suggestions. We plan to resubmit the manuscript reflecting the requested changes. Specifically, we will:

- We will detail more thoroughly the effects of agar volume on gene expression changes elicited by LW agar treatment.

- We will investigate whether the tensile stress introduced by hard agar is similar to soil compaction by an analysis with existing literature.

- Assess more rigorously the suitability of the ANCOVA model for assessing water potential changes of different media types.